# Assessment of quality of life in asthmatic children and adolescents: A cross sectional study in West Bank, Palestine

**Maher Khdour** [1] *, **Malek Abu Ghayyadeh**[1], **Dua'a Al-Hamed**[2], **Hussam Alzeerelhouseini** [1], **Heba Awadallah**[1]

**1** Faculty of Pharmacy, Al-Quds University, Abu Deis, West Bank, Palestine, **2** Pharmacy department Ramallah & Al-bireh Health Directorate, West Bank, Palestine

* mkhdour@staff.alquds.edu, maher.khdour@gmail.com

**Data Availability Statement:** All relevant data are within the manuscript and its Supporting information files.

## Abstract

### Background

Asthma is one of the most common chronic illnesses among children and adolescents. It can severely affect their quality of life (QoL). Our study assessed the QoL and analyzed potential risk factors for poor QoL among asthmatic children and adolescents.

### Methods

This was a cross-sectional comparative study. Pediatric Asthma Quality of Life Questionnaire (PAQLQ) was used to measure the QoL and Asthma Control Test (ACT) was used to evaluate asthma control. The Chi-square test and independent t-test were used to compare variables. We used Multivariate logistic regression to identify the association between determinants and outcomes. Statistical significance was set at p<0.05.

### Results

We recruited 132 participants. We found that 47 patients (35.6%) had controlled Asthma and 85 patients (64.3%) had uncontrolled Asthma. When compared to uncontrolled asthma individuals, participants with controlled asthma had improved QoL and scored significantly higher in the symptom domain (P = 0.002), activity domain (P = 0.004), emotional domain (P = 0.002), and overall PAQoL scores (P = 0.002). Hospital admission affects significantly all domains of PAQOL (*P*<0.05). Poor QoL was significantly associated with hospitalization for asthma (OR = 3.4; CI: 2.77–3.94, *P = 0.01*), disease severity (OR = 3.0; CI: 2.41–3.61, *P = 0.01*), uncontrolled asthma (OR = 2.88; CI: 2.21–3.41, *P = 0.019*), and male gender (OR = 2.55; CI: 1.88–2.91, *P = 0.02*).

### Conclusions

The results of the present study showed that in children and adolescents, uncontrolled asthma, disease severity, and previously hospitalized patients were associated with poor

**Funding:** The author(s) received no specific funding for this work.

**Competing interests:** The authors have declared that no competing interests exist.

QoL. These factors must be considered when planning a comprehensive care plan for a better quality of life.

## Introduction

Asthma is ranked as the 14th most disability condition in the world, with 14% prevalence among children, posing a significant disease burden [1]. The past three decades have seen increases in both prevalence and severity of childhood asthma in particular, as indicated by increased hospitalization rates and asthma mortality [2, 3]. Beyond the global burden of the condition, frequent and episodic asthma attacks limit the ability of children to participate in normal activities and thus significantly affect their quality of life.

Among Middle Eastern children aged 13–14 years, asthma prevalence was reported to be around 7.6%, with the lowest rate in Iran (0.7%) and the highest in Iraq (22.3%, specifically in Bagdad). Meanwhile, an intermediate prevalence was observed in West Bank, Palestine (3.8%), but recent episodes of wheezing were higher (8.9%). Similar findings were obtained from nearby countries, such as Jordan, where the prevalence was 4.1% for asthma, and 8.3% for wheezing [4].

A poorly controlled asthma in a growing child may have detrimental effects on their emotional, intellectual, and physical development [5]. From a quality of life (QoL) perspective, it's essential to direct management decisions to asthma control rather than its severity, ensuring the QoL improvement [6, 7]. A broad description of health-related QoL encompasses numerous dimensions, from general well-being and physical functioning or symptoms through mental health aspects such as cognitive, emotional functioning, and social well-being and functioning [8].

QoL assessment helps assess the burden of disease, offers new insights into the effects of risk factors, and is also a critical public health tool for evaluating health policy requirements, such as directing a strategic plan, decision-making concerning the utilization of funds, and assessing the efficacy of medical technology and public health initiatives. In assessing the QoL in children with Asthma, one of the most commonly used is the Pediatric Asthma Quality of Life Questionnaire (PAQoLQ), which provides insight into the patient's perspective, their experience of chronic illness, the comparison of procedures, drugs, and other interventions [9], highlighting QoL enhancement the main objective of asthma management [10–12]. Children with asthma have QoL as reduced as children with other chronic diseases such as nephrotic syndrome, chronic kidney disease, and epilepsy [13–15].

Several studies investigating asthma control in children are from abroad, and there is a lack of local studies that extensively explore the problem in Palestine. Therefore, this study aimed to assess the QoL and analyze potential risk factors for poor QoL among asthmatic children and adolescents.

## Methods

### Study design

This was a cross-sectional comparative study on children and adolescents diagnosed with asthma at two primary governmental outpatient clinics in Ramallah and Hebron, West Bank, Palestine, from November 2020 to March 2021.

## Study population and eligibility criteria

Asthmatic children and adolescents 6–17 years old who had been diagnosed by a physician at least 6 months prior were eligible for this study. A six-month period allowed adequate time to determine whether or not the children's asthma was controlled and not identify other comorbidities. Children without confirmed asthma diagnosis, asthmatic children with acute exacerbation at the time of recruitment, and children with asthma aged less than 6 years and more than 17 years were excluded.

## Sampling

We calculated a minimum sample size of 130 using a sample size calculator (see http://www.raosoft.com) to achieve a 5% margin of error, 95% confidence interval, and 80% response rate for 280 patients in the 2 clinics. We used a simple random sampling technique to recruit 132 participants.

## Data collection instrument

A structured, self-administered questionnaire was used for data collection. The questionnaire consisted of three sections: 1) demographic characteristics and clinical information, 2) the Asthma Control Test (ACT), and 3) the PAQLQ. Asthma control was evaluated using the ACT, a validated, internationally recognized asthma control assessment tool [15, 16]. Its questions use a 5-point Likert-type rating scale and assess both daytime and nocturnal asthma symptoms, use of rescue medications, and the effect of asthma on daily activities for the last four weeks. The scores range from 5 (poor control of asthma) to 25 (complete control of asthma), with higher scores reflecting greater asthma control [15]. The internal consistency reliability of the ACT survey was 0.85, measured by Cronbach's alpha. A score of <19 points indicates uncontrolled asthma, and $\geq$ 19 indicates controlled asthma.

Children's quality of life was measured using the PAQLQ, a validated self-reported questionnaire consisting of 23 questions spanning three domains (symptoms, activity limitation, and emotional function) [17]. Five questions evaluate the distress activities, ten questions are about the discomfort caused by asthma attacks, and eight questions inquire about the emotional function by assessing how asthma frustrates, scares, annoys, or upsets the patients [17, 18]. Responses are rated on a 7-point Likert scale ranging from 1 (most severe impairment) to 7 (no impairment at all). The arithmetic mean of the answers to the 23 questions is determined to give the total score [17, 18], for which a higher score indicates better quality of life. PAQoL score ranges from minimal or no impairment ($\geq$ 6.0) to severe impairment (< 3.0) [19]. PAQoL is reproducible in patients who were stable with an intraclass correlation coefficient (ICC = 0.95), which also indicates the instrument's strength to discriminate between subjects of different impairment levels.

## Data collection measures

The study investigator and clinical pharmacists working at the study sites recruited eligible children and adolescents and matched them with one main caregiver (father, mother, others) who accompanied them and consented, on their behalf, to participate in the study after receiving all information about the study aim and objectives. The participants completed the questionnaires in 15 to 20 min, with 80% completing the questionnaire in 15 min or less.

## Statistical analysis

Statistical analysis was performed using SPSS software (version 25; SPSS Inc., Chicago, IL, USA). The chi-squared test was used to measure the relationship between categorical variables, and the independent t-test to measure the association between the means of continuous variables. Descriptive statistics were performed using means and standard deviations for numerical data and as summary frequencies and percentages for categorical data. ANOVA and other Tests were used as appropriate. In statistical tests, $P$-values $\leq 0.05$ were considered to be statistically significant.

## Ethical approval

This study was approved by the Research Ethical Committee of Al-Quds University (Ref No: 164/REC/2020). Approval for data collection was obtained from the Palestinian Ministry of Health in Ramallah and Hebron, Palestine. Study details were provided to each patient and their parents (father or mother) with the information about the right to withdraw at any time. Caregivers gave written consent, and children gave verbal consent before data collection. Confidentiality was ensured by the anonymity of the questionnaires and no disclosure of information collected to anyone outside the study. The study was conducted following the Declaration of Helsinki.

## Results

### Participants' characteristics

A total of 132 children (6–17 years old) participated in the study. The mean age was 8.6 ± 3.0 years, and 79 (59.8%) were boys. Only one-third of the parents, either fathers or mothers, had a university education, 40 (30.3%) and 46 (34.8%), respectively. All participants' mean duration of asthma was 30.9 ± 19.4 months, and most of the parents were smokers, 101 (76.5%). Clinically most of the patients were taking Short-Acting Beta Agonists, 101(76.5%), and two-thirds were taking ICS, 88 (66.6%). More than half of the participants, 75 (56.8%), had a history of asthma-related hospital admissions (Table 1).

### Level of asthma control

We divided participants into two groups based on their ACT scores: controlled asthma (ACT score≥ 19) and uncontrolled (ACT score <19). Most (67%) participants with controlled asthma had mild asthma, 22% had moderate asthma, and 11% had severe asthma. In contrast, 53% of participants with uncontrolled asthma had mild asthma, 26 had moderate asthma, and 21% had severe asthma. Uncontrolled asthma was significantly related to lower parents' education levels (p = 0.04 for father education and p = 0.02 for mother's education), frequent hospital ED admissions (p = 0.001), smoking exposure at home (p = 0.001), school absenteeism (p = 0.001), and systemic steroids use (p = 0.001) (Table 1).

### Quality of life (QoL)

The most common restricted activity during the week preceding the study among patients was playing with friends (19.05%), followed by playing football (13%) and walking (12.1%) (Fig 1).

The overall PAQoL score was significantly higher in the controlled asthma group (Mean = 4.37) (p = 0.002) than in the uncontrolled asthma group (Mean = 3.56). The controlled asthma group also scored higher on all components indicating better symptoms control (*P* = 0.002), lower activity limitation (*P* = 0.004), and better emotional state (*P* = 0.002) compared to the uncontrolled asthma group (Table 2).

**Table 1. Patient characteristics and univariate analysis results.**

| Variable n (%) | All patients (132) | Controlled asthma (47) | Un-controlled asthma (85) | P-value |
|---|---|---|---|---|
| **Gender** | | | | |
| Male | 79(59.84) | 29(61.70) | 50(58.82) | 0.74[¥] |
| Female | 53(40.15) | 18(38.29) | 35(41.17) | |
| **Age** | | | | |
| 6–9 years | 85(64.39) | 32(68.08) | 53(62.35) | 0.77[¥] |
| 10–14 years | 35(26.51) | 13(27.65) | 23(27.05) | |
| 14–17 years | 12(9.09) | 3(6.38) | 9(10.58) | |
| **Father's education** | | | | |
| Illiterate/elementary | 24(18.18) | 5(10.63) | 19(22.35) | 0.04[¥] |
| School-level | 68(51.51) | 34(72.34) | 34(40) | |
| College/university level | 40(30.30) | 28(59.57) | 12(14.11) | |
| **Mother's education** | | | | |
| Illiterate/elementary | 26(19.69) | 4(8.51) | 22(25.88) | 0.02[¥] |
| School-level | 60(45.45) | 14(29.78) | 46(54.11) | |
| College/university level | 46(34.84) | 29(61.70) | 17(20) | |
| **BMI** | | | | |
| Underweight | 71(53.78) | 26(55.31) | 45(52.94) | |
| Normal | 46(34.84) | 14(29.78) | 32(37.64) | 0.45[¥] |
| Overweight | 10(7.57) | 6(12.76) | 4(4.70) | |
| Obese / $\geq 95^{th}$ percentile | 5(3.78) | 1(2.12) | 4(4.70) | |
| **ED visit last year** | | | | |
| 0 | 25(18.93) | 15(31.91) | 10(11.76) | <0.001[¥] |
| 1–2 | 75(56.81) | 25(53.19) | 50(58.82) | |
| $\geq 3$ | 32(24.24) | 7(14.89) | 25(29.41) | |
| **Smoking exposure at home** | | | | |
| Yes | 101(76.51) | 29(61.70) | 72(84.70) | 0.01[¥] |
| No | 31(23.48) | 18(38.29) | 13(15.29) | |
| **School absenteeism** | | | | |
| Yes | 89 (67.4) | 23 (48.9) | 66 (77.6) | 0.01[¥] |
| No | 43 (32.6) | 24 (51.1) | 19 (22.4) | |
| **Medications** | | | | |
| SABA | 101(76.51) | 36(76.59) | 65(76.47) | 0.53 |
| ICS | 88(66.66) | 30(63.82) | 55(64.70) | 0.41 |
| Nebulizers | 57(43.18) | 20(42.55) | 37(43.52) | 0.53 |
| Interleukin modifiers | 20(15.15) | 8(17.02) | 12(14.11) | - - |
| Anticholinergics | 16(12.12) | 8(17.02) | 8(9.41) | - - |
| Systemic steroids | 45(26.51) | 11(23.40) | 34 (40.0) | 0.01 |
| Duration of asthma (months ±SD) | 30.9 ± 19.4 | 30 ± 19.4 | 32.2 ± 20.1 | 0.6* |

Qol: Quality of Life.

[¥]: Chi-square test for categorical groups.

*: T-Student Test.

A significant difference was found in symptoms severity, activity impairment, and emotional function. Few participants in the controlled asthma group (4%) had severe symptoms compared to those in the uncontrolled asthma group (15%). Severe activity impairment was reported by the majority (66%) in the uncontrolled group, whereas, in the controlled group, it

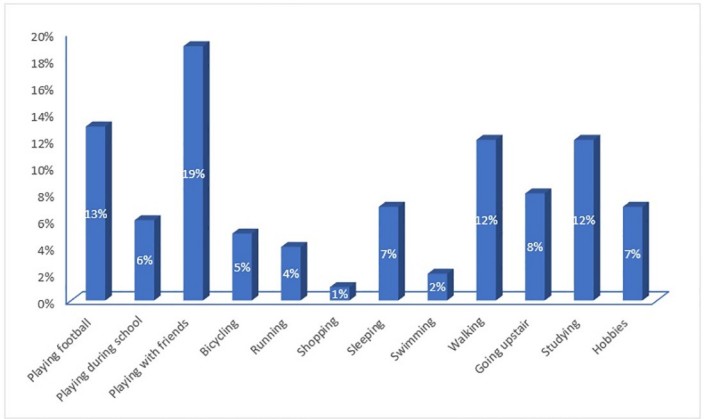

**Fig 1. Common activities affected by asthma.**

was reported by 40%. Moderate emotional impairment was higher in the uncontrolled asthma group (84%) than in the controlled asthma group (61%) (Fig 2).

Participants with no hospital admission history for asthma scored significantly higher (Mean 4.4) (p = = 0.001) in QoL than participants with a history of one or more hospital admissions (Mean = 3.1). Hospital admissions didn't significantly affect school attendances. However, school absenteeism was significantly associated with lower PAQoL symptoms (p = 0.01) and emotion function (p = 0.04) scores (Table 3).

Male gender, hospitalization for asthma, asthma severity, and control were significantly associated with poor QoL (p<0.05). The hospitalization had the strongest association with the QoL (OR = 3.4; CI: 2.77–3.94), followed by asthma severity (OR = 3.0; CI: 2.41–3.61), uncontrolled asthma (OR = 2.88; CI: 2.21–3.41) and male gender with the weakest association (OR = 2.55; CI: 1.88–2.91) (Table 4).

## Discussion

According to our knowledge, this is the first study to assess the QoL and associated factors among Palestinian children and adolescents with asthma. Moreover, our study took into account the asthmatic children and adolescents and their parents' reports. As such, its findings represent the quality of life from the participants' and their parents' perspectives.

**Table 2. Effect of asthma control on PAQoL scores.**

| The Dependent Variables | Asthma Control | N | Mean | Std. Deviation | t | P* |
|---|---|---|---|---|---|---|
| Overall PAQOLQ scores | Uncontrolled Asthma | 85 | 3.56 | 1.18 | -3.14 | .002 |
| | Controlled Asthma | 47 | 4.37 | 1.54 | | |
| Score of symptoms | Uncontrolled Asthma | 85 | 4.51 | 1.15 | -3.15 | .002 |
| | Controlled Asthma | 47 | 5.29 | 1.47 | | |
| Score of activity limitation | Uncontrolled Asthma | 85 | 2.68 | 1.26 | -2.96 | .004 |
| | Controlled Asthma | 47 | 3.52 | 1.72 | | |
| Score of emotional function | Uncontrolled Asthma | 85 | 4.24 | 1.27 | -3.17 | 002 |
| | Controlled Asthma | 47 | 5.07 | 1.51 | | |

* Independent Samples T-Test

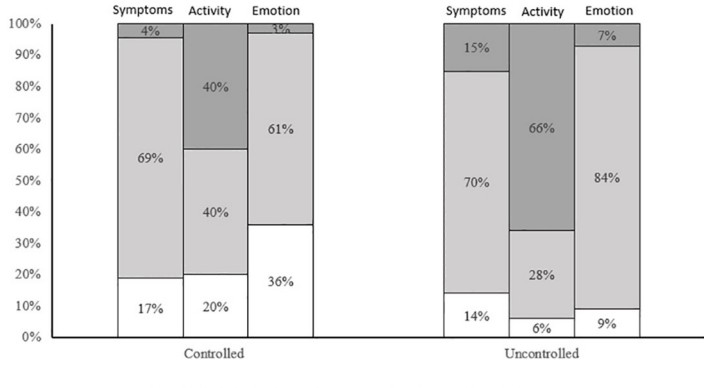

**Fig 2. The distribution of the degree of impairment reported by the patients in the PAQoL components (symptoms, activity, and emotional domains) by levels of asthma control.**

All three PAQoL domains (activity, symptoms, and emotional function) were affected by uncontrolled asthma, with activity being the most affected (p = 0.004). Similar findings were reported in different studies on Saudi asthmatic children and adolescents [20] and Swedish asthmatic children aged 7–9 years old [21]. In Nigeria, a survey on asthmatic children and adolescents revealed that participants were concerned about exacerbations and associated daily life activity limitations [22]. However, some other studies found the symptoms domain to be the most affected [22, 23]. This discrepancy could be attributed to the exclusion of patients with exacerbations or any related symptoms, which could impact the perceived QoL.

Only 47 (35.6%) of our study participants had controlled asthma, similar to the reported 30% of patients with asthma having controlled asthma status in a survey of 7236 asthmatic patients from the Middle East and North Africa [24]. Participants reported low quality of life, which had a direct association with disease severity and control. Asthma control is directly linked to a patient's physical and emotional capability and their QoL [25, 26]. Another study from Egypt revealed that asthmatic children and adolescents with controlled asthma had higher PAQoL scores than their counterparts with uncontrolled asthma [27]. Garina et al. conducted a study on Indonesian asthmatic adolescents aged 12–14 years old. They found correlations between the total PAQoL score and asthma severity (p<0.001, r = −0.5) and the level of asthma control (p<0.001, r = 0.6) [28].

**Table 3. Effect of hospital admission and school absenteeism on patients' QoL scores.**

| | School absenteeism | Mean ± SD | t-test | P | Hospital admission | Mean ± SD | t-test | p |
|---|---|---|---|---|---|---|---|---|
| Overall PAQoL scores | No | 4.1 ± 1.12 | 1.7 | 0.07 | No | 4.4 ± 1.92 | 3.1 | 0.01 |
| | Yes | 3.6 ± 1.13 | | | Yes | 3.1 ± 1.22 | | |
| Score of activity limitation | No | 3.2± 1.21 | 0.9 | 0.13 | No | 3.9 ± 1.18 | 3.3 | 0.01 |
| | Yes | 2.9 ± 1.14 | | | Yes | 2.6 ± 1.11 | | |
| Score of symptoms | No | 5.2 ± 1.17 | 2.4 | 0.01 | No | 4.3 ± 1.32 | 2.8 | 0.01 |
| | Yes | 4.5 ± 1.41 | | | Yes | 3.2 ± 1.24 | | |
| Score of emotional function | No | 4.9 ± 1.22 | 2.0 | 0.04 | No | 4.3 ±1.31 | 2.8 | 0.01 |
| | Yes | 3.9 ± 1.19 | | | Yes | 3.1 ± 1.17 | | |

PAQoL: Pediatric asthma quality of life; P < 0.05: Significant; SD: Standard deviation

**Table 4. Risk factors associated with poor QoL (Multiple regression analysis).**

| Variables in the Equation | | | | | |
|---|---|---|---|---|---|
| | **B** | **S.E.** | **OR** | **p-value** | **95% CI** |
| **Gender (Male)** | **0.93** | **0.381** | **2.55** | **0.022** | **1.88–2.91** |
| **Hospitalization** | **1.22** | **0.588** | **3.4** | **0.010** | **2.77–3.94** |
| Age | -0.193 | 0.439 | 0.825 | 0.661 | 0.11–1.21 |
| History of Asthma | 0.113 | 0.764 | 1.12 | 0.584 | 0.56–1.43 |
| **Asthma severity** | **1.09** | **0.54** | **3.0** | **0.011** | **2.41–3.61** |
| Duration of Asthma | 0.086 | 0.101 | 1.09 | 0.611 | 0.44–1.68 |
| Fathers' education | 0.166 | 0.451 | 1.18 | 0.580 | 0.71–2.01 |
| Mothers' education | 0.26 | 0.611 | 1.30 | 0.191 | 0.87–1.79 |
| Systemic Steroids | 0.111 | 0.199 | 1.11 | 0.24 | 0.73–1.94 |
| BMI | -0.127 | 0.439 | 0.88 | 0.677 | 0.43–1.46 |
| Smoking exposure | 0.190 | 0.494 | 1.21 | 0.211 | 0.69–2.10 |
| **Asthma control*** | **1.06** | **0.491** | **2.88** | **0.019** | **2.21–3.41** |

*Uncontrolled asthma, CI: Confident interval, OR: Odd Ratio. S.E: Standard error, B: the regression weight.

The observed decrease in all domains of the PAQoL score among children having asthma-related hospitalizations implies that, in the case of poor asthma control with more frequent symptoms, physical activity is low, and patients are more emotionally affected, hence the lower PAQoL scores [19, 29]. Similarly, a study of asthmatic patients conducted in Australia found a strong association between the number of hospital admissions and poor QoL [30]. Another study carried out in the USA using the Child Health Questionnaire Parental Form-28 identified strong associations between asthma severity and pediatric asthma with poor QoL [31]. Another indicator of poor asthma control in children is school absenteeism, which is associated with lower PAQoL scores, particularly in our study participants' PAQoL symptoms and emotional domains scores. These findings align with Dean et al. findings of a strong correlation between school absenteeism with poor QoL in the USA [32].

Multivariate analysis revealed the association between the male gender and poor QoL. Contrary to our findings, Indinnimeo et al. found that female patients were more likely to have poor QoL than male patients. He attributed that to a higher proportion of females in his study with exposure to secondary smoking [33]. Our study also indicated that exposure to secondary smoking was significantly related to more uncontrolled asthma cases (p = 0.001) (Table 1).

Our analysis did not identify increased BMI as having any significant effect on asthma control or PAQoL scores. Several previous studies reported lower QoL and asthma control scores among asthmatic children and adolescents with increased BMI [34–36]. Our study indicated no significant effect of higher BMI on asthma control or PAQoL scores. This might be due to a higher percentage (88.62% for underweight and normal weight) of patients with low BMI in our sample.

## Limitations

We recruited children with asthma who had attended primary care units, thereby excluding patients in hospitals with relatively more severe asthma. Consequently, our study sample may not represent all children with asthma, and our findings may not be generalized to the asthmatic population in Palestine at large.

Our study was questionnaire-based and relied on self-reports, which makes it prone to recall bias, as such overestimation or underestimation of asthma control or quality of life by participants.

This was a cross-sectional study. Therefore, causal associations cannot be drawn between the factors examined here. Prospective follow-up studies are recommended to confirm the results.

## Conclusion

This study highlighted that participants had poor QoL. The unique needs of asthmatic children and adolescents must be considered when planning a comprehensive care plan for a better quality of life, with particular emphasis given to uncontrolled asthma, disease severity, and previously hospitalized patients as there are the most prominent risk factors for poor QoL. Further research on the factors contributing to poor asthma control, the psychological effects of asthma, hospitalized patients, parents' quality of life, and the importance of screening for behavioral problems among asthmatic children is recommended.

## Supporting information

**S1 Table. Univariate and multiple regression analyses evaluating the association of quality of life and clinical variables.**
(DOCX)

## Author Contributions

**Conceptualization:** Maher Khdour, Malek Abu Ghayyadeh.

**Data curation:** Malek Abu Ghayyadeh, Dua'a Al-Hamed, Hussam Alzeerelhouseini, Heba Awadallah.

**Formal analysis:** Maher Khdour, Malek Abu Ghayyadeh.

**Methodology:** Dua'a Al-Hamed, Hussam Alzeerelhouseini.

**Software:** Heba Awadallah.

**Writing – original draft:** Maher Khdour, Malek Abu Ghayyadeh, Heba Awadallah.

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
