## [Decision Letter · Decision Letter 0]

27 Apr 2022

PONE-D-22-00022Assessment of quality of life in asthmatic children and adolescents: A Cross sectional study in West Bank, PalestinePLOS ONE

Dear Dr. Khdour,

Thank you for submitting your manuscript to PLOS ONE. After careful consideration, we feel that it has merit but does not fully meet PLOS ONE’s publication criteria as it currently stands. Therefore, we invite you to submit a revised version of the manuscript that addresses the points raised during the review process.

We look forward to receiving your revised manuscript.

Kind regards,

Tai-Heng Chen, M.D.

Academic Editor

PLOS ONE

Journal Requirements: 

2. You indicated that you had ethical approval for your study. In your Methods section, please ensure you have also stated whether you obtained consent from parents or guardians of the minors included in the study or whether the research ethics committee or IRB specifically waived the need for their consent

Reviewers' comments:

Reviewer's Responses to Questions

**Comments to the Author**

1. Is the manuscript technically sound, and do the data support the conclusions?

Reviewer #2: Partly

Reviewer #3: Yes

2. Has the statistical analysis been performed appropriately and rigorously? 

Reviewer #2: Yes

Reviewer #3: Yes

3. Have the authors made all data underlying the findings in their manuscript fully available?

Reviewer #2: No

Reviewer #3: Yes

4. Is the manuscript presented in an intelligible fashion and written in standard English?

Reviewer #2: No

Reviewer #3: Yes

5. Review Comments to the Author

Reviewer #2: General comments

The authors conducted a cross-sectional study whose aim was to investigate the quality of life (QOL) of children and adolescents with asthma. One hundred and thirty-two (132) participants were evaluated with the validated asthma control test (ACT) questionnaire and the pediatric asthma quality of life (PAQOL) questionnaire. The major finding was the low QOL scores seen among children with uncontrolled asthma. Although an interesting study (given the novelty in the authors’ clime), the authors need to address and clarify some major and minor issues on the manuscript for its improvement. For instance, the authors should state clearly the research question. Did they investigate if asthma control and severity affect QOL in children and adolescents with asthma? Or if there is a difference in the QOL between children with controlled and uncontrolled asthma?

Specific comments

Major issues

1. Abstract: For a manuscript of this nature (in line with the journal’s guidelines), I thought it should have been a structured abstract rather than an unstructured abstract. In line with the former, a background or introduction to the study should have been the first subheading. Under this subheading, the authors should have stated the research problem or gap they sought to solve or fill. I find the conclusion of the study findings rather curious. The authors stated: ‘Palestinian asthmatics children have surprisingly low quality of life, especially children with uncontrolled asthma.’ Were the authors not expecting children with uncontrolled asthma not to have low QOL? They should recall that they mentioned in the results that factors that affected QOL included disease severity, uncontrolled asthma etc.

2. Introduction: The first sentence needs referencing. The third sentence suggests (rightly so) that children with uncontrolled asthma have low QOL scores. Thus, I suggest the authors should mention previous studies on the QOL of children with uncontrolled asthma, and the QOL instruments used, the flaws (if any) on their use, and the advantage of using their chosen QOL instrument. These facts should form part of the justification of their present study. I do not agree with the statement before the study objective which read: ‘Although the treatment objectives for asthma are relative clear, the relationship between the asthma and the QOL of children is still not well-understood topic’ Rather, it is well documented that children with asthma has reduced QOL like children with other non-communicable chronic diseases such as nephrotic syndrome, chronic kidney disease and epilepsy.

3. Methods: The authors stated that the study was ‘a cross-sectional, analytical clinical.’ They should clarify the phrase. I would prefer to call the study ‘a cross-sectional comparative study’. It appears the authors compared the QOL of children with controlled asthma with that of children with uncontrolled asthma. Although the authors stated the sample size calculation and the inclusion/exclusion criteria, they failed to indicate their sampling method. The third item in their exclusion criteria were children who had difficulty in understanding the questionnaire. Why should it be so when the ACT questionnaire was administered by parental proxy? On the other hand, the PAQOL questionnaire was directly administered (self-administered) to the patients who were aged 6-17 years. Were the younger age groups able to understand the questions related to all the domains? Wouldn’t interviewer-administration have been a better option? Why was a non-generic health-related QOL questionnaire like PedsQLTM 4.0 Generic Core Scale not used as part of the study instruments? Given that asthma causes psychosocial disorders in children, this tool would have properly evaluated the psychosocial domain of the patients, which PAQOL could not achieve.

4. Results: This section was not clearly defined in the manuscript. I guess it started with the subheading titled ‘Patient characteristics’. The first sentence under the subheading is vague- ‘Of the 188 patients approached for this study, a total of 132 agreed to take part (response rate 70.2%).’ Was informed consent applicable to all the patients (stated age range of the patients was 6-17 years? The sentence also exposed the absence or the lack of clarity in the employed sampling method? The stated mean age was 8.6 ± 3.0. Was it in years or months? Table 2 appears redundant. I think the prose suffices. The title of Figure 1 should be modified for clarity. The subheading titled ‘The association between PAQOL, hospital admission and school absenteeism’ should be replaced by the title of Table 4 which aptly captures the discussion better. In Table 5, what is the meaning of the abbreviation ‘B’ and ‘Sig.’ for the variables predicting lower PAQOL scores?

5. Discussion: This section appears poorly written. In order to provide a robust discussion, I advise the authors to re-write this section using the following suggestions: (1) paragraph one should comprise a summary of the major research gap or problem they are trying to fill or address and its importance (2) paragraph two should provide a critical analysis of the major findings of the study and how they compare with previously published studies (3) paragraph three should discuss additional findings and how they fit with existing literature (4) paragraph four should be on study limitations (5) paragraph five can focus on future research directions (6) the last paragraph should be the overall conclusion and the major impact of this study (how does your study address the research question or fill the research gap?).

Minor issues

1. Several syntax and grammar errors litter throughout the manuscript. Given that the authors may not be from native English-speaking environment, I advise an English-language editing of the manuscript to make it polished.

Reviewer #3: Thank you to give me the chance to review the research paper “Assessment of quality of life in asthmatic children and adolescents: A Cross sectional study in West Bank, Palestine. The study is well organized and written, however, I have some comments which I hope the authors would consider to improve the quality of the manuscript.

Abstract

- Please rephrase the objective to become clearer

- In the abstract section No need to mention Inc., Chicago, IL, USA [optional]

- As the beginning of the paragraph please rephrase “ This indicated a better symptoms

- Control”

Introduction

- Could you add a paragraph to show the level of asthma control among pediatric in Palestine?

- As the last paragraph in the introduction of your study , rephrase the sentence to be fit with the aims in your study in Palestine

Methods

- In the ACT test would you mention the original validity and reliability of the questionnaire and NOT only at your setting

- Same to Qol questionnaire the original validity by the publisher

- Elaborate more, what was the average interview time or how long it took to complete the survey

- Were the pilot findings included in the study analysis? Not clear

Results

- You mentioned Number / % in some points you mentioned only %. Please be consistent

- In the text mention exactly the P-Values as expressed in the Table 4

- Also explain the numbers between brackets are percentages or SD

-

Discussion

- Rephrase “ Another indicator of less well-controlled disease is school absenteeism, which is likewise”

- The abbreviation of PQOL and Qol. please modify the manuscript accordingly

6. PLOS authors have the option to publish the peer review history of their article (what does this mean?). If published, this will include your full peer review and any attached files.

Reviewer #2: **Yes: **Samuel Uwaezuoke

Reviewer #3: No

---

## [Author Response · Author response to Decision Letter 0]

3 May 2022

Response to editor and reviewer comments was uploaded in this submersion

---

## [Decision Letter · Decision Letter 1]

30 May 2022

PONE-D-22-00022R1Assessment of quality of life in asthmatic children and adolescents: A Cross sectional study in West Bank, PalestinePLOS ONE

Dear Dr. Khdour,

Thank you for submitting your manuscript to PLOS ONE. After careful consideration, we feel that it has merit but does not fully meet PLOS ONE’s publication criteria as it currently stands. Therefore, we invite you to submit a revised version of the manuscript that addresses the points raised during the review process.

We look forward to receiving your revised manuscript.

Kind regards,

Tai-Heng Chen, M.D.

Academic Editor

PLOS ONE

Journal Requirements:

Reviewers' comments:

Reviewer's Responses to Questions

**Comments to the Author**

1. If the authors have adequately addressed your comments raised in a previous round of review and you feel that this manuscript is now acceptable for publication, you may indicate that here to bypass the “Comments to the Author” section, enter your conflict of interest statement in the “Confidential to Editor” section, and submit your "Accept" recommendation.

Reviewer #1: (No Response)

Reviewer #2: All comments have been addressed

2. Is the manuscript technically sound, and do the data support the conclusions?

Reviewer #1: Yes

Reviewer #2: Yes

3. Has the statistical analysis been performed appropriately and rigorously? 

Reviewer #1: Yes

Reviewer #2: Yes

4. Have the authors made all data underlying the findings in their manuscript fully available?

Reviewer #1: Yes

Reviewer #2: Yes

5. Is the manuscript presented in an intelligible fashion and written in standard English?

Reviewer #1: Yes

Reviewer #2: Yes

6. Review Comments to the Author

Reviewer #1: After reframing the abstract,objectives, results and the discussion the study provides a valid conclusion that may help in planning a compressive care program for pediatric asthmatic patients.

Reviewer #2: You have substantially addressed my concerns about your manuscript. However, painstakingly go through your abstract section again to ensure that the information there is clear and aligns with the body of your manuscript. For instance, the following statement lack clarity- 'The participants with controlled asthma significantly score higher in symptoms control (P= 0.002), activity (P=0.004), emotional state (P=0.002) and in all over PAQoL scores (P=0.002) among control group.' The control group mentioned here, I guess, refers to uncontrolled asthma group. The 'cross-sectional analytic' study under Methods should align with 'Cross-sectional comparative' study mentioned in the body of the manuscript

7. PLOS authors have the option to publish the peer review history of their article (what does this mean?). If published, this will include your full peer review and any attached files.

Reviewer #1: No

Reviewer #2: **Yes: **Samuel N Uwaezuoke

---

## [Author Response · Author response to Decision Letter 1]

31 May 2022

Revision 2

Journal Requirements:

Due to major revision in the first manuscript, 3 references were retracted and replaced with new references in revision 1. However, now all references are relevant and cited in the main text. 

Reviewer #1: After reframing the abstract, objectives, results, and the discussion the study provides a valid conclusion that may help in planning a compressive care program for pediatric asthmatic patients.

Thank you 

Reviewer #2: You have substantially addressed my concerns about your manuscript. However, painstakingly go through your abstract section again to ensure that the information there is clear and aligns with the body of your manuscript. For instance, the following statement lack clarity- 'The participants with controlled asthma significantly score higher in symptoms control (P= 0.002), activity (P=0.004), emotional state (P=0.002) and in all over PAQoL scores (P=0.002) among control group.' The control group mentioned here, I guess, refers to uncontrolled asthma group.

The sentence was amended and clear. Controlled asthma (mean better controlled) and score higher (better quality of life domains) 

When compared to uncontrolled asthma individuals, participants with controlled asthma had improved QoL and scored significantly higher in the symptom domain (P=0.002), activity domain (P=0.004), emotional domain (P=0.002), and overall PAQoL scores (P=0.002) 

The 'cross-sectional analytic' study under Methods should align with 'Cross-sectional comparative' study mentioned in the body of the manuscript

The sentence was amended accordingly

---

## [Decision Letter · Decision Letter 2]

15 Jun 2022

Assessment of quality of life in asthmatic children and adolescents: A Cross sectional study in West Bank, Palestine

PONE-D-22-00022R2

Dear Dr. Khdour,

We’re pleased to inform you that your manuscript has been judged scientifically suitable for publication and will be formally accepted for publication once it meets all outstanding technical requirements.

Kind regards,

Tai-Heng Chen, M.D.

Academic Editor

PLOS ONE

Reviewers' comments:

Reviewer's Responses to Questions

**Comments to the Author**

1. If the authors have adequately addressed your comments raised in a previous round of review and you feel that this manuscript is now acceptable for publication, you may indicate that here to bypass the “Comments to the Author” section, enter your conflict of interest statement in the “Confidential to Editor” section, and submit your "Accept" recommendation.

Reviewer #1: All comments have been addressed

Reviewer #2: All comments have been addressed

2. Is the manuscript technically sound, and do the data support the conclusions?

Reviewer #1: Yes

Reviewer #2: Yes

3. Has the statistical analysis been performed appropriately and rigorously? 

Reviewer #1: Yes

Reviewer #2: Yes

4. Have the authors made all data underlying the findings in their manuscript fully available?

Reviewer #1: Yes

Reviewer #2: Yes

5. Is the manuscript presented in an intelligible fashion and written in standard English?

Reviewer #1: Yes

Reviewer #2: Yes

6. Review Comments to the Author

Reviewer #1: The authors have addressed all the concerns and have made relevant changes to the manuscript. These amendments have increased the clarity of the abstract.

Reviewer #2: (No Response)

7. PLOS authors have the option to publish the peer review history of their article (what does this mean?). If published, this will include your full peer review and any attached files.

Reviewer #1: No

Reviewer #2: **Yes: **Samuel Uwaezuoke

---

## [Editor Report · Acceptance letter]

20 Jun 2022

PONE-D-22-00022R2 

Assessment of quality of life in asthmatic children and adolescents: A Cross sectional study in West Bank, Palestine 

Dear Dr. Khdour:

I'm pleased to inform you that your manuscript has been deemed suitable for publication in PLOS ONE. Congratulations! Your manuscript is now with our production department. 

Kind regards, 

on behalf of

Dr. Tai-Heng Chen 

Academic Editor

PLOS ONE